# Semi-Automatic MRI Muscle Volumetry to Diagnose and Monitor Hereditary and Acquired Polyneuropathies

**DOI:** 10.3390/brainsci11020202

**Published:** 2021-02-06

**Authors:** Friederike S. Bähr, Burkhard Gess, Madlaine Müller, Sandro Romanzetti, Michael Gadermayr, Christiane Kuhl, Sven Nebelung, Jörg B. Schulz, Maike F. Dohrn

**Affiliations:** 1Department of Neurology, Medical Faculty of the RWTH Aachen University, 52074 Aachen, Germany; fbaehr@ukaachen.de (F.S.B.); burkhard.gess@rwth-aachen.de (B.G.); madlaine.mueller@rwth-aachen.de (M.M.); sromanzetti@ukaachen.de (S.R.); jschulz@ukaachen.de (J.B.S.); 2Department of Neurology, Inselspital Bern, CH-3010 Bern, Switzerland; 3Institute of Imaging and Computer Vision, RWTH Aachen University, 52074 Aachen, Germany; michael.gadermayr@fh-salzburg.ac.at; 4Salzburg University of Applied Sciences, 5020 Salzburg, Austria; 5Department of Diagnostic and Interventional Radiology, Medical Faculty of the RWTH Aachen University, 52074 Aachen, Germany; ckuhl@ukaachen.de (C.K.); snebelung@ukaachen.de (S.N.); 6Department of Diagnostic and Interventional Radiology, Medical Faculty, University Düsseldorf, 40225 Düsseldorf, Germany; 7JARA-BRAIN Institute Molecular Neuroscience and Neuroimaging, ForschungszentrumJülich GmbH and RWTH Aachen University, 52425 Jülich, Germany; 8Dr. John T. Macdonald Foundation, Department of Human Genetics and John P. Hussman Institute for Human Genomics, Miller School of Medicine, University of Miami, Miami, FL 33136, USA

**Keywords:** muscle volume, neuropathy, CIDP, CMT, biomarker, MRI

## Abstract

With emerging treatment approaches, it is crucial to correctly diagnose and monitor hereditary and acquired polyneuropathies. This study aimed to assess the validity and accuracy of magnet resonance imaging (MRI)-based muscle volumetry.Using semi-automatic segmentations of upper- and lower leg muscles based on whole-body MRI and axial T1-weighted turbo spin-echo sequences, we compared and correlated muscle volumes, and clinical and neurophysiological parameters in demyelinating Charcot-Marie-Tooth disease (CMT) (*n* = 13), chronic inflammatory demyelinating polyneuropathy (CIDP) (*n* = 27), and other neuropathy (*n* = 17) patients.The muscle volumes of lower legs correlated with foot dorsiflexion strength (*p* < 0.0001), CMT Neuropathy Score 2 (*p* < 0.0001), early gait disorders (*p* = 0.0486), and in CIDP patients with tibial nerve conduction velocities (*p* = 0.0092). Lower (*p* = 0.0218) and upper (*p* = 0.0342) leg muscles were significantly larger in CIDP compared to CMT patients. At one-year follow-up (*n* = 15), leg muscle volumes showed no significant decrease.MRI muscle volumetry is a promising method to differentiate and characterize neuropathies in clinical practice.

## 1. Introduction

Polyneuropathies are manifold in etiology and manifestations. With an overall prevalence of 1:2500 [1], Charcot-Marie-Tooth disease (CMT) is the most common hereditary neuropathy, constituting a group of rare diseases associated with about 100 different genes to date [2]. Chronic inflammatory demyelinating polyneuropathy (CIDP) is an acquired disease of autoimmune origin that can potentially lead to comparably severe disability if untreated. With a prevalence of 2–3:100.000 [3], CIDP is even less frequent than CMT. As such, both entities belong to the group of rare diseases and therefore remain challenging to diagnose [4]. With new upcoming treatment approaches for both hereditary [5,6] and inflammatory [7,8,9,10] neuropathies, it becomes more and more important to recognize underlying causes and monitor treatment response [1,11]. Currently, the standard diagnostics comprise a detailed neurological examination and nerve conduction studies (NCS) [12]. The former is dependent on both the patient’s collaboration and the examiner’s experience, while the latter is associated with the unpleasant experience of electric stimulation. Subsequently, there is a clinical need for accurate, reproducible, and non-invasive diagnostic tools that may help to differentiate these disease entities and monitor their clinical course.

In this work, we focused on a new method of diagnosing and monitoring polyneuropathies: semi-automatic muscle segmentation. Muscle MRI has been gaining importance in identifying and locating pathological changes in neuromuscular disorders [13,14]. The freely available software ITK-SNAP [15,16] is mainly used in cerebral volumetry, including the delineation of tumor areas [17] or the illustration of atrophy patterns [18]. As a proof of concept, we have previously shown that employing the software for muscle segmentation is an accurate method to distinguish myopathy from polyneuropathy patients [19].

In the present study on MRI muscle volumetry, we further examined a larger neuropathy cohort, emphasizing three main issues: (i) does muscle volume correlate with clinical and electrophysiological parameters? (ii)Are there detectable differences between the etiological subgroups including acquired and hereditary polyneuropathies? (iii)What are the longitudinal changes in muscle atrophy patterns? Accordingly, we hypothesized that muscle volumes correlate with clinical and neurophysiological measures, differ significantly comparing acquired and hereditary neuropathies, and reflect the longitudinal disease course over time.

Clinical examinations and nerve conduction studies as well as genetic analyses (if CMT is suspected) are indispensable elements for the diagnostics of polyneuropathies to date. The intention of this work is not to replace these methods, but to provide an additive tool that is objective and non-invasive, and potentially complements and facilitates the diagnostic process.

## 2. Materials and Methods

### 2.1. Patient Selection

Patients were recruited at the Neuromuscular Outpatient Clinic, Department of Neurology of the RWTH Aachen University Hospital, Aachen, Germany, from October 2014 to August 2019. The eligibility criteria were a confirmed neuropathy diagnosis and an age of majority. In patients with hereditary neuropathies, the molecular genetic diagnoses were confirmed prior to study inclusion. The acquired neuropathies were ascertained following clinical and electrophysiological standards [12].

Muscle MRI studies were prospectively performed in 46 patients within the frame of diagnostic standards. The indication was always made in a clinical context in order to monitor muscle atrophy progression. Clinical and NCS data were assessed in close temporal relation to each patient’s muscle MRI. Twenty-six additional datasets (MRI, NCS, and clinical data) were created retrospectively from the clinic’s digital archive. In 15 of the total 57 patients, we were able to compile two longitudinal datasets. The mean interval between MRI and clinical examination was two months. All examinations and analyses were performed and supervised by the same, experienced examiners. Patients with skeletal anomalies (e.g., amputations), potentially interfering primary diseases (e.g., tumors), missing informed consent, or incomplete MRI datasets were excluded. For the direct comparison of muscle volumes in demyelinating neuropathies, the CIDP cohort consisted exclusively of those patients with a typical CIDP, not including individuals with pure sensory subtypes.

### 2.2. Clinical Examination

A detailed clinical examination was performed in all patients. The assessment of muscle strength of upper and lower limbs (range from zero to five according to medical research council (MRC) [20]) was included, as well as a precise assessment of the patients’ gait patterns, including steppage and afferent ataxia. A detailed inspection of the undressed extremities focused on muscle asymmetries and atrophy, the presence or absence of pes cavus and claw toes, and on wound-healing disorders or amputations. Abnormalities in deep tendon reflexes, particularly Achilles tendon reflexes, patellar tendon reflex, and biceps tendon reflex, were accurately documented. Moreover, we focused on the qualitative assessment of all sensory modalities, including the perception of light touch, pinprick, vibration, position, and temperature, with their respective distribution patterns, namely, discontinuous or length-dependent, and the exact levels of perception. Pallesthesia was examined at the malleolar and wrist levels using a 64 Hz RydelSeiffer tuning fork with a default scale from zero to eight. Clinical outcomes were summarized using the CMT Neuropathy Score Version 2 (CMTNS-2), a composite score comprising sensory and motor symptoms, pallesthesia, muscle strength, and NCS (distal compound muscle action potential of one ulnar and sensory nerve action potential of one radial nerve, both on the non-dominant side) [21,22].

### 2.3. Nerve Conduction Studies

To assess the quality (axonal or demyelinating) and distribution pattern (length-dependence, symmetry, and motor or sensory predominance) of the neuropathy, NCSs were conducted by the same team of practiced examiners in all patients. The standard parameters to be assessed at the tibial, median, and/or ulnar nerve were nerve conduction velocity (NCV), compound motor action potentials (CMAP), distal motor latency (DML), and F-waves. Sensory nerve action potentials (SNAP) and sensory NCV were orthodromically measured for the sural nerve on both sides, and antidromically for the radial nerve on one side.

### 2.4. MRI Studies

All patients underwent a whole-body muscle MRI, which was performed at the Department of Radiology of the RWTH Aachen University Hospital, Aachen, Germany. MRI examinations were performed on two clinical 1.5T MRI scanners, i.e., Philips Achieva or Philips Ingenia (Philips Healthcare, Best, The Netherlands), using the built-in volume transmitter/quadrature detection receiver coil (Q body coil) or the dStreamWholeBody coil, respectively. The imaging protocol consisted of axial T1-weighted (T1w) turbo echo spin (TSE) and axial short tau inversion recovery (STIR) sequences. For the T1w TSE sequences, the parameters were as follows: Achieva: repetition time (TR) = 902 ms; echo time (TE) = 17 ms; TSE factor 6; flip angle (FA) = 90°; field of view (FOV) = 529 × 529 mm; Acquisition Matrix (AM) = 436 × 348 pixels; pixel size 1.2 × 1.5 mm/pixel; slice thickness (ST) = 6 mm; slice gap (SG) = 7 mm; number of signal averages (NSA) = 2; Ingenia: TR = 620 ms; TE = 9 ms; TSE factor 7; FA = 90°; FOV = 542 × 542 mm; AM = 399 × 308 pixels; pixel size 1.4 × 1.8 mm/pixel; ST = 5 mm; SG = 6.2 mm; NSA = 1. For the STIR sequences, the parameters were as follows: Achieva: TR = 3972 ms; TE = 64 ms; TSE factor 30; FA = 90°; FOV = 500 × 500 mm; AM = 336 × 252 pixels; pixel size 1.5 × 2.0 mm/pixel; ST = 6 mm; SG = 7 mm; NSA = 2; Ingenia: TR = 2147 ms; TE = 60 ms; TSE factor 17; FA = 90°; FOV = 500 × 500 mm; AM = 384 × 297 pixels; pixel size 1.3 × 1.7 mm/pixel; ST = 5 mm; SG = 6.2 mm; NSA = 1.

### 2.5. Image Analysis

Data were pseudonymized and converted from DICOM (digital imaging and communications in medicine) to NIFTI (neuroimaging informatics technology initiative) format using the ITK-SNAP software [15,16,23]. For measuring each patient’s upper and lower leg muscle volumes, the segmentation was semi-automatically performed using the T1w axial images. The active contour algorithm, which is implemented in the ITK-SNAP software, constitutes the automatic part of the procedure, while preparation and post-processing were done by the investigator.

The region of interest (ROI) for the thighs was defined in the craniocaudal dimension from the femoral head to the superior pole of the patella. For the lower legs, the ROI was selected from the medial tibial condyle to the inferior articular surface of the tibia. We used the thresholding pre-segmentation mode based on an upper and a lower threshold. The thresholds were set at individual levels for each image, so that the contrast between muscles and other tissues was highlighted best each time. Following the determination of certain image intensities, i.e., gray-scale values, one or more size-adjustable spherical “seeds” were placed within the muscle during the step of “initialization”. Starting out from those seed regions, the active contour iteratively converged towards the aspired 3D muscle volume. The implemented muscles of the lower legs were extensors, superficial and deep flexors, as well as peroneus muscles. The segmentation of thigh muscles included sartorius muscle, quadriceps muscle, hamstring muscles, and adductors.

The finally obtained segmentation of the muscles was re-evaluated and manually corrected if necessary. As an example of typical errors in male patients, the active contour algorithm falsely recognized the scrotal muscle and included it into the segmentation requiring manual removal. On average, the segmentation of one patient (upper and lower left leg) amounted to about 20 to 60 min, depending on the need for manual adjustment. The segmentation process is exemplarily illustrated in Appendix A.

### 2.6. Statistics

Statistical analyses were conducted with Graphpad Prism 5. We used the Student’s *t*-test to compare the parametric features of one group with another. Gaussian distribution was ensured by running the D’Agostino and Pearson omnibus normality test. Linear regression analyses were done to evaluate clinical, paraclinical, and score correlations. Correlations were quantified using Pearson’s correlation coefficient. Only the latest examination results and images of each patient were included with the regular calculations. To maintain statistical independence, only the left lower extremities were analyzed, by default. Data are given as means ± standard deviation. The level of statistical significance was defined as α = 0.05.

## 3. Results

In our study, 57 neuropathy patients were included in total. The age ranged from 28 to 85 years (mean age, 60.2 ± 12.3 years). The study population consisted of 42 men and 15 women. Since CIDP is more frequent in men than in women [24], this imbalance in sexes is representative. Thirty patients were diagnosed with autoimmune neuropathies, such as CIDP (mean age, 66.2 ± 8.8 years), or CIDP-variants such as multifocal motor neuropathy (MMN) or multifocal acquired demyelinating sensory and motor neuropathy (MADSAM). We further included 13 patients with demyelinating CMT (mean age, 48.5 ± 13.8 years) and 7 patients with other hereditary neuropathies, including axonal CMT (*n* = 2), hereditary transthyretin-related systemic amyloidosis (ATTRv) with polyneuropathy (*n* = 2), hereditary neuropathy with liability to pressure palsies (HNPP) (*n* = 1), hereditary myoneuropathy (*n* = 1), and distal hereditary motor neuropathy (dHMN) (*n* = 1). Another seven patients were classified as neuropathies of other (diabetic neuropathy (*n* = 2), chronic idiopathic axonal polyneuropathy (CIAP) (*n* = 2), Guillain–Barrésyndrome (GBS) residual (*n* = 1)) or unknown (*n* = 2) origin.

In general, the most common pathogenic variant associated with demyelinating CMT is the heterozygous *PMP22* duplication [1], which was also the most prevalent mutation in our hereditary neuropathy sub-cohort (*n* = 6). The exact distribution of all etiologies and mutations present in this study, as well as a more detailed description of the entire study cohort, is given in Figure 1 and Table 1.

### 3.1. Clinical Description of the CIDP and Demyelinating CMT Collective

The two most representative patient sub-cohorts were the ones withCIDP and demyelinating CMT, which we will herein describe and compare in more detail. An overview on the clinical and paraclinical examination results is given in Appendix A.

The mean age of disease onset among the CIDP patients was 58.9 ± 9.6 years, with a resulting mean duration of disease of 7.3 ± 4.6 years. The most frequently mentioned first symptoms were neuropathic pain and muscle weakness (41% each), but paresthesia was frequently noted as well (37%). Most CIDP patients (63%) described their disease progression as stable, 82% of whom received standard treatment with intravenous immunoglobulins (IVIG) during the term of the study. Alongside paresthesias(83%), CIDP patients most frequently stated muscle cramps (67%) and neuropathic pain (46%) as the leading current symptoms. Fine motor skills were impaired in 40%, and almost 67% admittedto have a limited walking distance, which was clinically attributed to impaired heel walking in 73%, to steppage gait in 52% of the cohort, and to impaired walking on tiptoes in 52%. Walking aids were required in 28% in the CIDP cohort. Jerk reflexes were reduced or absent in 96%. While nearly all CIDP patients had a reduced pallesthesia at the lower limbs (96%), still 20% likewise showed a reduced perception of vibration in the upper limbs. The pinprick perception was impaired in 74% (lower limbs). Regarding muscle strength, foot dorsiflexion, plantarflexion, and hip flexionwere most frequently affected. In summary, these outcomes led to an average CMTNS-2 of 12.0 ± 5.2 for the CIDP cohort (though the CMTNS-2 is in principle a CMT severity score).

In comparison, the cohort of demyelinating CMT patients had a mean age of disease onset of only 11.5 ± 7.1 years.Their mean disease duration summed up to 37.1 ± 12.5 years at the time of the examination. As opposed to the CIDP cohort, the CMT patients mostly reported gait disorders as their first symptom (58%), and a slowly deteriorating disease progression was stated in 50%. Their most frequently mentioned current symptoms were paresthesia (75%) and neuropathic pain (70%), and almost 67% noted their fine motor skills to be affected. Up to 75% of the CMT patients affirmed that they had a limited walking distance, and 31% depended on at least one walking aid. Almost the entire CMT cohort revealed severe difficulties with heel walking (92%), but only 62% showed affected toe walking. Achilles tendon reflexes were reduced or absent in 77%. With 92%, the perception of light touch of the lower limbs was more frequently impaired than in CIDP, as was temperature sensation (92%). At the lower limbs, the perception of vibration (69%) and pinprick were frequently diminished as well (83%). Regarding muscle strength, foot dorsiflexion, foot eversion and inversion were particularly affected. The CMT cohort achieved a mean CMTNS-2 of 15.9 ± 4.8 in total.

For all sensory parameters (CIDP and CMT), the lower limbs were consistently more affected than the upper limbs.

### 3.2. Muscle Volumes

Compared to our CIDP cohort (1442.6 cm^3^ ± 501.2 cm^3^), the demyelinating CMT patients’ (1050.1 cm^3^ ± 451.9 cm^3^) lower leg muscle volumes were significantly lower (*p* = 0.0218) (Figure 2a). The mean thigh volumes were also significantly greater in CIDP (typical CIDP only) patients (4423.6 cm^3^ ± 1207.7 cm^3^) compared to CMT patients (3565.2 cm^3^ ± 1041.3 cm^3^) (*p* = 0.0342) (Figure 2b). In all chronic neuropathy patients, the muscle volumes of the lower legs correlated with foot dorsiflexion strength (*p* < 0.0001) (Figure 2c). As a marker of overall disease severity, the lower leg muscle volumes correlated inversely with the CMTNS-2 score (*p* < 0.0001) (Figure 2d). Gait instability as the first symptom of disease onset was a significant predictor of smaller lower leg muscle volumes in all patients (*p* = 0.0486). There was no significant correlation between the upper or lower leg muscle volumes and sensory parameters such as vibration perception.

In the CIDP cohort, the patients’ lower leg muscle volume correlated with the NCV of the left tibial nerve, so that patients with slower conduction velocities had lower muscle volumes as well (*p* = 0.0092) (Figure 2e). Meanwhile, other neurophysiological parameters, including DML and CMAP, did not correlate with muscle volumes. As the tibial nerve NCV, DML, and CMAP values were not representatively evaluable in the CMT cohort, which was related to the already advanced nerve damage, this correlation analysis could not be performed in this subgroup. Instead, we studied the median, ulnar, and radial nerve. However, no significant correlations could be detected with upper and lower leg muscle volumes (e.g., linear regression analysis of CMT patients’ lower leg muscle volumes and median nerve conduction velocity: *p* = 0.2723).

Considering the total study population, there was no significant correlation between muscle volume and disease duration, although a tendency towards decreasing volumes with increasing disease duration was indicated (*p* = 0.0578 for lower legs). Comparing lower leg muscle volumes in our overall cohort, the difference between males and females was not significant (*p* = 0.0580).

Exemplary segmentation results of the lower legs of a mildly affected CIDP patient and a severely affected CMT patient are shown in Figure 3.

### 3.3. Longitudinal Data

In 15 longitudinally examined patients, we compared baseline and follow-up data with an interval of at least twelve months (mean time interval 21.6 ± 8.4 months). Their lower leg muscle volumes showed no significant decrease over time. The clinical parameters did not significantly differ over time, and neither did the NCS (CMTNS-2, *p* = 0.1446). Ten patients had CIDP, one had demyelinating CMT, and four patients had other neuropathy diagnoses (1× CIAP, 1×dHMN, 1× GBS-residual, 1× axonal CMT). Nine patients with CIDP were treated with immunoglobulins at the time of at least one assessment.

## 4. Discussion

In this study, we showed that leg muscle volumes correlate with clinical and nerve conduction parameters, and differ significantly between CIDP and CMT patients, representing acquired and hereditary polyneuropathies.

We conclude that MRI muscle volumetry, a non-invasive imaging method, can be combined with semiautomatic post-processing routines to systematically assess atrophy patterns in chronic neuropathies. As muscle volumes significantly differed between CMT and CIDP patients and correlated with the CMTNS-2, which is a combined clinical and neurophysiological score, we conclude that the method is a potential tool for future diagnostic routines.

Hereditary and acquired neuropathies comprise a large group of variable rare diseases that all merit prompt diagnostic and therapeutic considerations in order to achieve the best possible outcomes. In the past years, the number of available therapies for different types of neuropathies has grown immensely, making it even more urgent to define new biomarkers for clinical trials and practice. In this study, 82% of the CIDP patients were effectively treated with intravenous immunoglobulins. These and subcutaneous immunoglobulins have become the accepted or even preferred alternatives over steroids in the treatment of CIDP for years now [8,25]. For CMT1A, represented by six patients in this cohort, the novel substance PXT003 containing a fixed combination of baclofen, naltrexone, and sorbitol has already undergone a phase three clinical trial with promising results [5], and the decision on approval is currently pending. With patisiran and inotersen, two new substances for the treatment of ATTRv amyloidosis have been approved in 2018 as another example of recent developments in pathophysiology-based treatment in hereditary neuropathies [6].

In this work, we applied the MRI muscle volumetry technique in a well-characterized patient collective with rare diseases, including CMT and CIDP (prevalences: CMT 1:2500 [1], CIDP 2–3:100.000 [3]). Comparable, representative cohort sizes, continuity in the study personnel and protocol, and the breadth of the assessed parameters are therefore considered the major strengths of this study.Because the method of semiautomatic muscle segmentation was already established in our previous study in healthy subjects [19], we did not repeat MRI examinations in healthy controls.

Throughout the study population, there was no significant correlation between disease duration and muscle volume (*p* = 0.0578 for lower legs). However, patients with longer disease durations tended to have smaller muscle volumes, as expected, since neuropathies are usually progressive diseases and can lead to massive muscle atrophies. We recognize that for comparison between the subgroups, a matching age and disease duration would be favorable. Considering that CMT patients typically have an earlier onset and slower disease course than CIDP patients, the herein described differences do not reflect the ideal study design, but the real-life situation in clinical practice. As another potential limitation, the study collectives were not matched to their body size and weight, which might have been another confounder. A previous study has, however, shown that MRI muscle volumes do not significantly correlate with the body mass index in patients with neuromuscular diseases [19].

The longitudinal analysis of 15 patients in an interval of at least twelve months showed no significant decrease in leg muscle volumes, which was different from what was expected. This could be due to the fact that in the group of CIDP patients, nine out of ten patients received immunoglobulins and the other etiologic subgroups were too small to be meaningful. The small number of longitudinally examined patients (*n* = 15) is, however, a potential limitation. Further investigation is needed to determine whether the method might be a valid marker in monitoring therapeutic response in CIDP or CMT patients.

Regarding clinical parameters, lower leg muscle volumes correlated significantly with foot dorsiflexion strength, but did not correlate with sensory parameters, including distal vibration perception. This is an expected result pointing towards the putative validity of the method, as both muscle volume and strength represent the motor nervous system, whereas vibration does not.

Lower leg muscle volumes correlated with NCV, but not with the CMAP of the tibial nerves. As most patients in our study had a demyelinating neuropathy, this correlation fits with the expectations, although there are currently scarcely any data in the literature that provide evidence that NCV correlates with muscle atrophy. Therefore, it cannot be excluded that this is an incidental finding, which needs to be reassessed in a larger cohort. In CMT patients, NCS performed at the lower limbs were mostly not representative due to advanced nerve damage. This reveals that such a standard method has weaknesses in this respect, while there is no such limitation for MRI analyses. Depending on the respective genes and mutations, even the demyelinating CMT subgroups vary distinctly in their specific electrophysiological patterns [26], which makes it even more difficult to draw representative conclusions from a heterogeneous cohort. Due to the rareness of every single CMT sub-entity, however, the individual subgroups were too small to meet the statistical requirements for a valid analysis here.

The typical atrophy pattern of CMT is distally dominant [27], whereas typical CIDP is known for its symmetrical affection of proximal as well as distal muscles [28]. In our collective, however, the thigh muscle volumes were larger in CIDP compared to CMT patients (*p* = 0.0342). As a possible explanation, CMT has a long progressive course with ascending muscle atrophies, such that these atrophies might outweigh the low-grade proximal atrophies of the CIDP patients in our cohort.

Previous studies have already shown that exploring neuromuscular imaging methods provides helpful information about the severity and patterns of muscle involvement [29]. In particular, hereditary neuropathies appear to be a rewarding subject of such investigations [30].

MRI is neither painful, (which NCSs are perceived to be by many patients), nor as dependent on the examiner’s experience or the patients’ collaboration as clinical evaluations are. However, this method excludes patients with (relative) contraindications for MRI, such as metal implants, cardiac pacemakers, or claustrophobia.

Since neuromuscular disorders regularly emerge to a highly variable degree and manifestation, gathering data of whole muscles promises more precise outcomes than analyzing just a few or even single slices, as has been done in previous studies [31,32,33]. Therefore, we consider using the ITK-SNAP software and creating 3Dfigures of whole muscles a clear improvement in the analysis of MRI data. The software is easy to handle and freely available [15,16].

Since the fatty degeneration of muscles usually increases concomitantly with disease progression [11], assessing the muscle fat fraction in neuropathy patients is another promising approach [31,34,35,36]. Whereas the segmentation method used in this study does not allow for the calculation of fat fractions, we are currently working on a protocol for the semi-autonomic segmentation of both muscle volume and fat fraction to overcome this issue.

It remains to consider that whole-body MRI studies are costly and dependent on sufficient technical and financial resources, which makes unstinting applications more difficult. However, the routine use of MRI has massively increased in the past years [37], and is predicted to grow further. Recent technological advances in the field of whole-body MRI, such as moving table platforms and smart coil integration within the magnet’s isocenter, have helped to streamline whole-body MRI studies by overcoming table range limitations and the need for constant patient repositioning.

As potential prospects, larger and more in-depth studies on semi-automatic segmentations of MRI datasets are needed. With the further follow-up and expansion of the study cohort, longitudinal data might provide interesting information on disease course and treatment response. Very rare diseases, such as ATTRv amyloidosis, which were just sparsely represented in our cohort, might also need further investigation with larger case numbers.

## 5. Conclusions

Both CIDP and CMT are progressive, disabling diseases, which can cause major impairment in participation and quality of life. This includes skeletal deformities, muscle weakness, sensory loss, and neuropathic pain. Since new therapeutic options are coming up by now, it is essential to determine the correct diagnosis as early as possible and to precisely monitor the disease course. Since NCS fail in dealing with advanced disease stages and clinical examinations are more dependent on the examiner, semiautomatic muscle MRI volumetry is an attractive additive method for future clinical practice. This study aimed at validating MRI muscle volumetry as a new and reliable method for polyneuropathy diagnostics, by correlating it with the current gold standards, including clinical examinations and NCS. We conclude that semiautomatic MRI muscle volumetry is a valid and accurate tool to help in diagnosing and distinguishing hereditary and acquired polyneuropathies; however, the clinical and electrophysiological examination still represents an indispensable and irreplaceable tool in the instrumental diagnosis of polyneuropathy.

## Figures and Tables

**Figure 1 brainsci-11-00202-f001:**
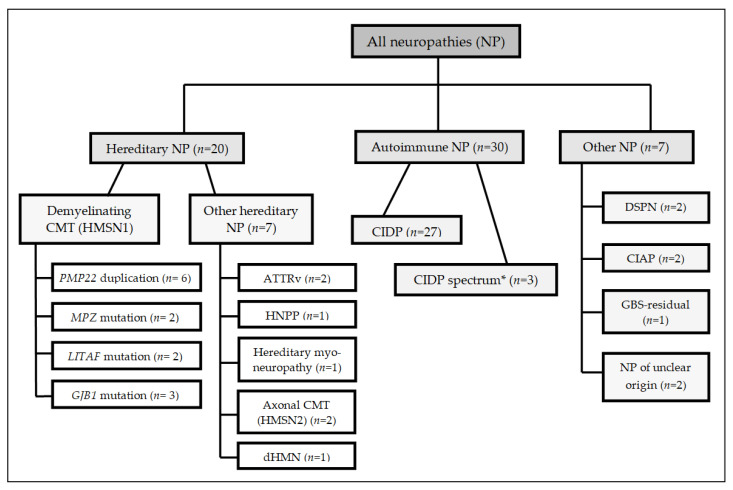
Organigram showing the etiologic classification of our study cohort and the particular numbers of each sub-collective. All mutations were heterozygous except for two male patients with hemizygous variants in *GJB1*. * CIDP spectrum includes atypical CIDP and variants such as MMN or MADSAM. Abbreviations in Figure 1: ATTRv, hereditary transthyretin-related systemic amyloidosis; CIAP, chronic idiopathic axonal polyneuropathy; CIDP, chronic inflammatory demyelinating polyneuropathy; CMT, Charcot-Marie-Tooth disease; dHMN, distal hereditary motor neuropathy; DSPN, diabetic distal symmetric polyneuropathy; GBS, Guillain–Barrésyndrome; HMSN, hereditary motor and sensory neuropathy; HNPP, hereditary neuropathy with pressure palsies; MADSAM, multifocal acquired demyelinating sensory and motor neuropathy; MMN, multifocal motor neuropathy; NP, neuropathy. *PMP22, MPZ, LITAF*, and *GJB1*: Genes that are associated with CMT.

**Figure 2 brainsci-11-00202-f002:**
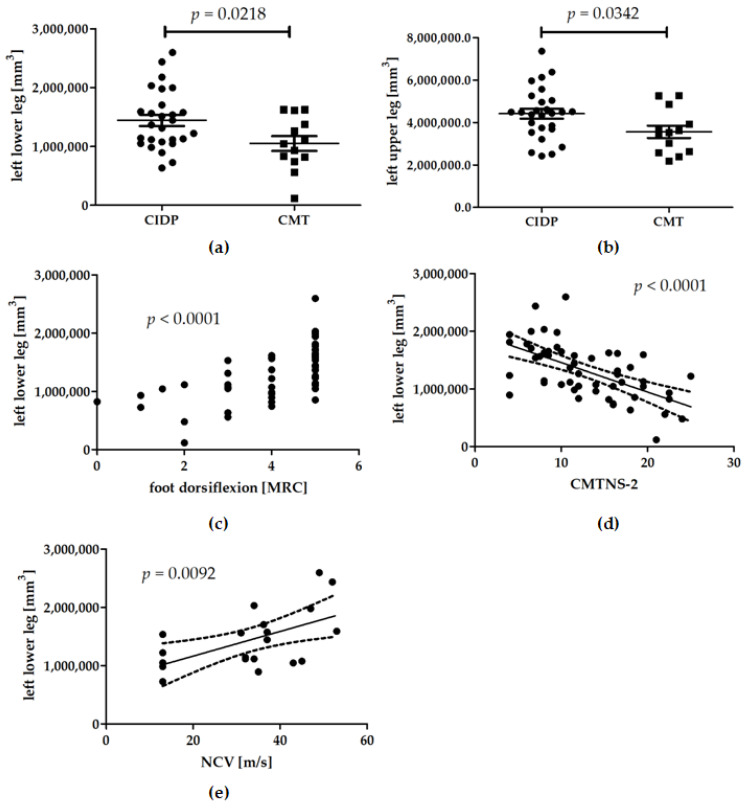
(**a**–**e**). MRI-based muscle volumetric data as a function of disease entity (**a**,**b**) and clinical as well as neurophysiological reference measures (**c**–**e**). (**a**,**b**) Comparison of left leg muscle volumes in CIDP (chronic inflammatory demyelinating polyneuropathy) vs. demyelinating CMT (Charcot-Marie-Tooth) patients: CIDP patients showed significantly higher muscles volumes in lower ((**a**), *p* = 0.0218) and upper ((**b**), *p* = 0.0342) legs than CMT patients. (**c**) Linear regression of left lower leg muscle volumes and foot dorsiflexion strength (range from zero to five according to the Medical Research Council (MRC)) (*p* < 0.0001). (**d**) Linear regression of left lower leg muscle volumes and Charcot-Marie-Tooth neuropathy score version 2 (CMTNS-2) (*p* < 0.0001). (**e**) Linear regression of the left lower leg muscle volumes, and CIDP patients’ nerve conduction velocity (NCV) of thetibial nerve (*p* = 0.0092). In four patients, the NCV was too slow to be detectable due to massively damaged nerves, and was therefore set one value below the lowest detectable NCV value.

**Figure 3 brainsci-11-00202-f003:**
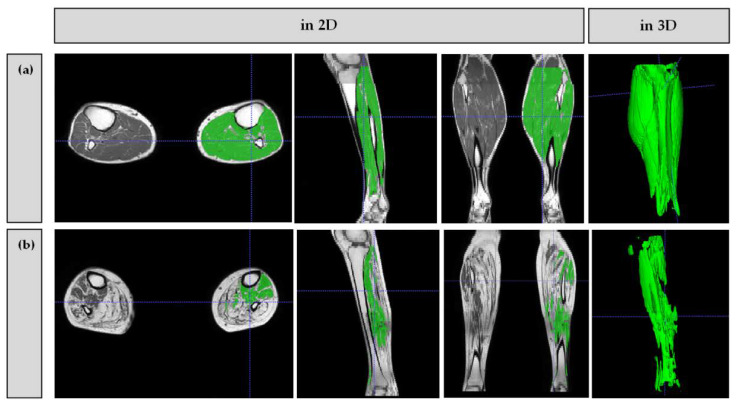
Exemplary segmentation results (left lower leg) of a mildy affected CIDP patient ((**a**) muscle volume 2030.2 cm^3^) and a severely affected CMT patient ((**b**) muscle volume 478.4 cm^3^).

**Table 1 brainsci-11-00202-t001:** Synopsis of the study population and means.

Diagnosis	*n*	*n* ♂	*n* ♀	Age (Years)	Muscle Volume Left Thigh (cm^3^)	Muscle Volume Left Lower Leg (cm^3^)	CMTNS-2 (x/36)	Foot Dorsiflexion Strength (x/5 by MRC)	Pallesthesia (Malleolar) (x/8)
Typical CIDP	27	23	4	66.2 ± 8.8	4423.6 ± 1207.7	1442.6 ± 501.2	12.0 ± 5.2	5 ± 0.7	3.0 ± 2.2
CIDP spectrum *	3	2	1	54.7 ± 4.2	4291.3 ± 1626.9	1461.5 ± 437.8	10.0 ± 4.0	5 ± 0.3	2.0 ± 2.8
CMT (demyelinating)	13	6	7	48.5 ± 13.8	3565.2 ± 1041.3	1050.1 ± 451.9	15.9 ± 4.8	4 ± 0.9	2.7 ± 2.3
*PMP22*duplication	6	3	3	49.5 ± 11.3	3189.2 ± 1140.2	877.9 ± 484.6	18.5 ± 2.9	3 ± 1.2	3.3 ± 1.8
*MPZ* mutation	2	1	1	53.0 ± 24.0	4355.3 ± 1289.7	1443.2 ± 253.9	10.0 ± 2.8	5 ± 0.0	5.8 ± 1.1
*LITAF* mutation	2	0	2	43.5 ± 21.9	3767.6 ± 216.4	1366.4 ± 365.4	11.8 ± 5.3	5 ± 0.0	5.3 ± 2.3
*GJB1* mutation	3	2	1	47.0 ± 15.4	3655.7 ± 1139.6	921.5 ± 413.3	17.3 ± 5.0	4 ± 0.3	1.7 ± 2.5
ATTRv amyloidosis	2	2	0	62.5 ± 13.4	3467.8 ± 87.6	1053.2 ± 282.7	17.5 ± 1.4	5 ± 0.0	0.0 ± 0.0
Other hereditary NP	5	4	1	51.8 ± 7.8	4780.8 ± 752.5	1390.6 ± 545.9	12.0 ± 7.5	4 ± 1.0	4.1 ± 2.6
DSPN	2	2	0	68.0 ± 11.3	4253.4 ± 2341.6	1477.2 ± 343.6	6.8 ± 3.9	5 ± 0.0	3.0 ± 2.8
CIAP	2	1	1	61.5 ± 3.5	4412.5 ± 1447.7	1362.9 ± 408.9	11.3 ± 3.9	4.5 ± 0.5	0.0 ± 0.0
GBS-residual	1	0	1	76.0	2296.2	824.4	22.5	0	2.5
Unclear NP	2	1	1	64.5 ± 6.4	3752.2 ± 1628.9	1545.8 ± 377.1	4.0	5 ± 0.0	4.0 ± 0.0
All patients	57	42	15	60.2 ± 12.3	4151.4 ± 1210.7	1327.0 ± 482.6	12.9 ± 5.6	5 ± 0.9	2.8 ± 2.2

* (includes atypical CIDP, MMN and MADSAM). Note: Data are presented as mean ± standard deviation, except for the foot dorsiflexion strength, which is presented as median ± mean deviation from the median. Abbreviations in Table 1: ATTRv, hereditary transthyretin-related systemic amyloidosis; CIAP, chronic idiopathic axonal polyneuropathy; CIDP, chronic inflammatory demyelinating polyneuropathy; CMT, Charcot-Marie-Tooth disease; CMTNS-2, Charcot-Marie-Tooth Neuropathy Score version 2; DSPN, diabetic distal symmetric polyneuropathy; GBS, Guillain–Barrésyndrome; MADSAM, multifocal acquired demyelinating sensory and motor neuropathy; MMN, multifocal motor neuropathy; MRC, medical research council; NP, neuropathy.

## Data Availability

The analyzed data are not publically available (following local and legal data protection guidelines).

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
