# Peer review of "Semi-Automatic MRI Muscle Volumetry to Diagnose and Monitor Hereditary and Acquired Polyneuropathies"

_brainsci, 2021, doi:10.3390/brainsci11020202_

Round 1
Reviewer 1 Report
This paper looks very interesting, however in my opinion Authorsshould modify the last part ( "We conclude that semiautomatic MRI muscle volumetry is a valid and accurate tool to help diagnosing and distinguishing hereditary and acquired polyneuropathies").
It would be better to write "...semiautomatic MRI muscle volumetric is a valid and accurate tool to help diagnosing and distinguishing hereditary
and acquired polyneuropathies , however the electromyographic examination
still represents the indispensable and irreplaceable tool in the instrumental diagnosis of polyneuropathy".
Author Response
Response to Reviewer 1 Comments
Point 1:
This paper looks very interesting, however in my opinion Authors
should modify the last part ("We conclude that semiautomatic MRI muscle volumetry is a valid and accurate tool to help diagnosing and distinguishing hereditary and acquired polyneuropathies"). It would be better to write "...semiautomatic MRI muscle volumetric is a valid and accurate tool to help diagnosing and distinguishing hereditary and acquired polyneuropathies, however the electromyographic examination still represents the indispensable and irreplaceable tool in the instrumental diagnosis of polyneuropathy".
Response to point 1:
Thank you very much for reviewing our manuscript! We fully agree that electromyography and electroneurography are indispensable elements in the diagnostics of polyneuropathies as well as the clinical examination, which is why we state the MRI volumetry as an “additive method” in our conclusion. Therefore, we are happy to change the final sentence of the conclusion as you suggested: “We conclude that semiautomatic MRI muscle volumetry is a valid and accurate tool to help diagnosing and distinguishing hereditary and acquired polyneuropathies, however the electromyographic examination still represents an indispensable and irreplaceable tool in the instrumental diagnosis of polyneuropathy” (page 10, lines 436-439)
We hope that our manuscript will find acceptance for publication in Brain Sciences.
Sincerely yours,
Friederike S. Bähr
Reviewer 2 Report
In this study, the validity and accuracy of semi-automatic MRI-based muscle volumetry were evaluated among patients with hereditary and acquired polyneuropathies. Muscle volumes of lower extremities were correlated with some clinical and electrophysiological parameters in such patients. It was also shown that leg muscles were significantly larger in CIDP compared to CMT patients.
Design and results are acceptable, but the interpretation should be reconsidered.
Even though a significant difference is seen in the leg muscle volumes between CIDP and CMT patients, differences in not only the disease duration but also the sex ratio significantly affect the result.
CMT is distal-dominant polyneuropathy.
One of the features of typical CIDP is the involvement of both proximal and distal muscles.
If upper leg volume were smaller in CIDP patients than in CMT patients, it would be interesting. However, the result was the opposite.
Please state your hypothesis about the result.
If you cannot present a reasonable reason, please move two graphs showing differences in muscle volumes between CIDP and CMT patients in the lower part of Fig. 2.
Does CIDP mean typical CIDP?
If CIDP group included DADS and pure sensory CIDP, please show them in the manuscript or supplemental materials.
There is no information about the height and weight of participants.
How do you think about the correction of muscle volumes according to the size of the body?
Author Response
Response to Reviewer 2 Comments
Point 1:
In this study, the validity and accuracy of semi-automatic MRI-based muscle volumetry were evaluated among patients with hereditary and acquired polyneuropathies. Muscle volumes of lower extremities were correlated with some clinical and electrophysiological parameters in such patients. It was also shown that leg muscles were significantly larger in CIDP compared to CMT patients. Design and results are acceptable, but the interpretation should be reconsidered. Even though a significant difference is seen in the leg muscle volumes between CIDP and CMT patients, differences in not only the disease duration but also the sex ratio significantly affect the result.
Response to point 1:
Thank you very much for reviewing our manuscript!
Thank you for pointing out this important issue! Disease duration and sex ratio are indeed aspects that may influence the results. In our study, however, neither aspect had a statistically significant influence. Additionally to the passages about disease duration (page 7, lines 305-307 and 9, lines 382-385), we therefore added the following aspect to our manuscript: “Comparing lower leg muscle volumes in our overall cohort, the difference between males and females was not significant (p = 0.0580).” (page 7, lines 282-284)
Point 2:
CMT is distal-dominant polyneuropathy.
One of the features of typical CIDP is the involvement of both proximal and distal muscles.
If upper leg volume were smaller in CIDP patients than in CMT patients, it would be interesting. However, the result was the opposite.
Please state your hypothesis about the result.
If you cannot present a reasonable reason, please move two graphs showing differences in muscle volumes between CIDP and CMT patients in the lower part of Fig. 2.
Response to point 2:
This is a very good point, which to address, we added the following section to the discussion part: “The typical atrophy pattern of CMT is distally dominant[27], whereas typical CIDP is known for its symmetrical affection of proximal as well as distal muscles[28]. In our collective, however, the thigh muscle volumes were larger in CIDP compared to CMT patients (p = 0.0342). As a possible explanation, CMT has a long progressive course with ascending muscle atrophies, so that these atrophies might outweigh the low-grade proximal atrophies of the CIDP patients in our cohort.” (page 9, lines 387-392)
Point 3:
Does CIDP mean typical CIDP? If CIDP group included DADS and pure sensory CIDP, please show them in the manuscript or supplemental materials.
Response to point 3:
Thank you for pointing this out. Yes, CIDP means typical CIDP. To clarify this, we added/changed the following sentences to our manuscript:
“For the direct comparison of muscle volumes in demyelinating neuropathies, the CIDP cohort consisted exclusively of those patients with a typical CIDP, not including individuals with pure sensory subtypes.” (page 2, lines 88-91)
“Mean thigh volumes were also significantly greater in CIDP (typical CIDP only) patients (4423.6 cm3 ± 1207.7 cm3) compared to CMT patients (3565.2 cm3 ± 1041.3 cm3) (p = 0.0342) (Figure 2b).” (page 6, lines 261-262)
We also changed the corresponding row header in Table 1 from “CIDP” to “Typical CIDP” (page 4).
Point 4:
There is no information about the height and weight of participants.
How do you think about the correction of muscle volumes according to the size of the body?
Response to point 4:
We fully agree that height and weight are interesting parameters that should be taken into consideration. To address this, we added the following sentence: “As another potential limitation, the study collectives were not matched to their body size and weight, which might have been another confounder. A previous study has, however, shown that MRI muscle volumes do not significantly correlate with the body mass index in patients with neuromuscular diseases [19].” (page 9, lines 357-361)
We hope that our manuscript will find acceptance for publication in Brain Sciences.
Sincerely yours,
Friederike S. Bähr
Reviewer 3 Report
The authors described the usefulness of MRI muscle volumetry in the diagnosis and evaluation of disease severity in hereditary and acquired polyneuropathies. Such tool evaluating muscle volume is interesting and might be useful. However, in this study, it seemed that the authors did not show apparent novel findings.
In introduction section,
Refs 5 to 7 were the articles about TTR-familial amyloid neuropathy. I cannot find ref 8 on Pubmed. TTR-FAP and CMTs are quite different conditions in the development of symptom and pathophysiologies. These references are not appropriate in this section.
CMT and CIDP have the quite different clinical course and symptom. In general, the two conditions can be easily distinguished by clinical examination. Muscle volume on MRI does not show function such as muscle strength, sensory disturbance, reflex, or nerve conduction speed. In addition, today, the diagnosis of accurate CMT is made by genetic tests and genetic information will be essential for the treatments of individual CMTs. MRI is just an additional information. The authors should re-write the introduction section.
Minor: How about the prevalence of CMT?
In method section,
The vibration sense was shown as 0 to 8. Did the authors use Rydel Seiffer tuning fork for the examination of vibration sense? Please clarify.
In result section,
The authors use the MRC score for the statistical analysis. MRC score is a "non-linear" scale. The mean MRC score that the authors used is inappropriate for the statistical analysis.
In the muscle volumetry analysis, CIDP showed higher volume than CMTs. This results was nothing special. The patients with lower muscle volume had lower foot dorsiflexion strength. This is also as expected.
The correlation between muscle volume and NCV is theoretically incidental phenomenon. Although fastest fibers loss is associated mild conduction slowing, NCV in demyelinating neuropathy is not associated with muscle atrophy. The authors should discuss above concern.
In CMTs, NCV, DML and CMAP in the median, ulnar, and radial nerves did not show significant correlation with lower leg muscle volume. In this study, "demyelinating CMTs" included CMT1A, CMT1B, CMT1c, CMTX1. Individual CMTs have specific conduction speed. For example, CMT1a show around 20m/s conduction velocity in upper limbs. In contrast CMTX1 shows intermediate conduction slowing around 35-40m/s. Conduction speed in CMTs does not change until disappearance of CMAP. I cannot understand what the authors would like to show.
In discussion section,
In introduction and discussion sections, the authors described ATTR amyloidosis repeatedly. What was the significance in the result in ATTR amyloidosis in this study? Did the MRI muscle volumetry show the specific pattern in ATTR amyloidosis, or show the usefullness in evaluating disease severity?
Author Response
Response to Reviewer 3 Comments
Point 1:
Refs 5 to 7 were the articles about TTR-familial amyloid neuropathy. I cannot find ref 8 on Pubmed.
Response to point 1:
Thank you very much for reviewing our manuscript! We are sorry that you cannot find reference number eight on Pubmed yet. This reference is a conference contribution, which we chose to cite, because the publication of the original paper is still pending. The cited conference paper can be found under the following link: Young, P.; Attarian, S.; Youcef, B.; Rinaudo, P.;... - Google Scholar
Point 2:
TTR-FAP and CMTs are quite different conditions in the development of symptom and pathophysiologies. These references are not appropriate in this section.
Response to point 2:
We fully agree that TTR-FAP and CMT have quite different symptoms and pathophysiologies. Our intention to mention it, though, was to point out that a lot of research is currently being done in the field of mechanism-based polyneuropathy therapy, which reinforces the need for appropriate diagnostic tools and outcome parameters. TTR-FAP is an example of the rapid development of new therapeutic options. Nevertheless, we have to admit that three references on this topic are too many, since the actual focus of the project is clearly on CIDP and CMT, so we decided to replace them with one very recent summary review that summarizes these information more concisely (new reference number 6): Dohrn, M.F.; Ihne, S.; Hegenbart, U.; Medina, J.; Züchner, S.L.; Coelho, T.; Hahn, K. Targeting transthyretin - Mechanism-based treatment approaches and future perspectives in hereditary amyloidosis. J Neurochem 2020, 10.1111/jnc.15233, doi:10.1111/jnc.15233.
Point 3:
CMT and CIDP have the quite different clinical course and symptom. In general, the two conditions can be easily distinguished by clinical examination. Muscle volume on MRI does not show function such as muscle strength, sensory disturbance, reflex, or nerve conduction speed. In addition, today, the diagnosis of accurate CMT is made by genetic tests and genetic information will be essential for the treatments of individual CMTs. MRI is just an additional information. The authors should re-write the introduction section.
Response to point 3:
We fully agree that genetic analyses are indispensable for the diagnosis of CMT and so are electrophysiology and clinical examination. To emphasize more clearly that muscle volumetry is intended only as a complementary method to facilitate the diagnosis of polyneuropathies, we have added the following section: “Clinical examinations and nerve conduction studies as well as genetic analyses (if CMT is suspected) are indispensable elements for the diagnostics of polyneuropathies to date. The intention of this work is not to replace these methods, but to provide an additive tool that is objective and non-invasive and potentially complements and facilitates the diagnostic process.” (page 2, lines 66-70)
Point 4:
Minor: How about the prevalence of CMT?
Response to point 4:
We absolutely agree that the rareness of the diseases discussed in this paper is important to address, therefore we decided to add the prevalences in our discussion part as follows: “In this work, we applied the MRI muscle volumetry technique in a well-characterized patient collective with rare diseases, including CMT and CIDP (prevalences: CMT 1:2500 [1], CIDP 2-3:100.000[3]).” (page 9, lines 345-346)
Point 5:
The vibration sense was shown as 0 to 8. Did the authors use Rydel Seiffer tuning fork for the examination of vibration sense? Please clarify.
Response to point 5:
Thank you for addressing this important point. We did use a Rydel Seiffer tuning fork for the examination of vibration sense and therefore added the information as follows: “Pallesthesia was examined at malleolar and wrist levels using a 64 Hz Rydel Seiffer tuning fork with a default scale from zero to eight.” (page 3, lines 106-107)
Point 6:
The authors use the MRC score for the statistical analysis. MRC score is a "non-linear" scale. The mean MRC score that the authors used is inappropriate for the statistical analysis.
Response to point 6:
Thank you for this important comment! To correct this, we now performed the calculations using the medians of the MRC values of the different subgroups and accordingly adjusted table 1 (page 4, line 196 – page 5, line 207) and the following passages: “Assessing muscle strength, foot dorsiflexion (mean 4.2 (MRC)), plantarflexion (mean 4.7 (MRC)), and hip flexion (mean 4.7 (MRC)) were mostly affected.” (page 6, lines 235-237)
“Regarding muscle strength, foot dorsiflexion (mean 3.6 (MRC)), foot eversion (mean 4.1 (MRC)), and inversion (mean 4.3 (MRC)) were particularly affected.” (page 6, lines 252-254)
Point 7:
In the muscle volumetry analysis, CIDP showed higher volume than CMTs. This results was nothing special. The patients with lower muscle volume had lower foot dorsiflexion strength. This is also as expected.
Response to point 7:
We completely agree that the main results were as expected. To further emphasize that this work is a validation study and thus requires correlation of the new method with clinically established gold standards, we have added the following passage: “This study aimed at validating MRI muscle volumetry as a new and reliable method for polyneuropathy diagnostics by correlating it with current gold standards, including clinical examinations and NCS.” (page 10, lines 429-431)
Point 8:
The correlation between muscle volume and NCV is theoretically incidental phenomenon. Although fastest fibers loss is associated mild conduction slowing, NCV in demyelinating neuropathy is not associated with muscle atrophy. The authors should discuss above concern.
Response to point 8:
To address your very good point, we adapted the following section: “Lower leg muscle volumes correlated with NCV, but not with CMAP of the tibial nerves. As most patients in our study had a demyelinating neuropathy, this correlation fits to the expectations, although there is currently scarcely any data in the literature that provides evidence that NCV correlates with muscle atrophy. Therefore, it cannot be excluded that this is an incidental finding, which needs to be reassessed in a larger cohort.” (page 9, lines 374-377)
Point 9:
In CMTs, NCV, DML and CMAP in the median, ulnar, and radial nerves did not show significant correlation with lower leg muscle volume. In this study, "demyelinating CMTs" included CMT1A, CMT1B, CMT1c, CMTX1. Individual CMTs have specific conduction speed. For example, CMT1a show around 20m/s conduction velocity in upper limbs. In contrast CMTX1 shows intermediate conduction slowing around 35-40m/s. Conduction speed in CMTs does not change until disappearance of CMAP. I cannot understand what the authors would like to show.
Response to point 9:
We fully agree that different subgroups of Charcot-Marie-Tooth disease differ in their specific conduction speeds. To better clarify this, we have added the following passage: “Depending on the respective genes and mutations, even the demyelinating CMT subgroups vary distinctly in their specific electrophysiological patterns [26], which makes it even more difficult to draw representative conclusions from a heterogeneous cohort. Due to the rareness of every single CMT sub-entity, however, the individual subgroups were too small to meet the statistical requirements for a valid analysis here.” (page 9, lines 380-384)
Point 10:
In introduction and discussion sections, the authors described ATTR amyloidosis repeatedly. What was the significance in the result in ATTR amyloidosis in this study? Did the MRI muscle volumetry show the specific pattern in ATTR amyloidosis, or show the usefullness in evaluating disease severity?
Response to point 10:
We completely agree that there is too much focus on TTR-amyloidosis here. In addition to the reference adjustment (as already mentioned in author’s answer point 2), we have adjusted the following section:
“The small transthyretin (TTR) stabilizing molecule tafamidis has been eligible in Europe for patients with ATTRv amyloidosis since 2011 (and in the United States of America since 2018) and offers a promising alternative to liver transplantation[7,28]. With patisiran, based on RNA interference, and inotersen, an antisense oligonucleotide, which both lead to knockdown of the TTR-protein by sequence specific silencing of TTR mRNA, another two new substances have been approved in 2018 making ATTRv amyloidosis a model disease for pathophysiology-based treatment in hereditary neuropathies[5,6]. With patisiran and inotersen, two new substances for the treatment of ATTRv amyloidosis have been approved in 2018 as another example of recent developments in pathophysiology-based treatment in hereditary neuropathies.” (page 8, line 332 - page 9, 342)
,
We hope that our manuscript will find acceptance for publication in Brain Sciences.
Sincerely yours,
Friederike S. Bähr
Round 2
Reviewer 2 Report
I believe the manuscript has been significantly improved.
Reviewer 3 Report
The manuscript was improved and I understood the usefulness of MRI volumetry in the evaluation of muscle atrophy in CMT and CIDP. I think we can use this method for the evaluation of longterm outcome of the treatment in CMTs or CIDP near future. However, I still have doubt about the usefulness for the differential diagnosis. Various factors, such as types of the disease, disease duration, genetic factors and etc, can influence the muscle atrophy in individual patients. For this reason, further investigation should be designed.